# Difficulties in Addressing Diagnostic, Treatment and Support Needs in Individuals with Intellectual Disability and Persistent Challenging Behaviours: A Descriptive File Study of Referrals to an Expertise Centre

**DOI:** 10.3390/ijerph20146365

**Published:** 2023-07-14

**Authors:** Gerda de Kuijper, Tryntsje Fokkema, Martine Jansen, Pieter J. Hoekstra, Annelies de Bildt

**Affiliations:** 1GGZ Drenthe/Department Centre for intellectual Disability and Mental Health, Middenweg 19, 9404 LL Assen, The Netherlands; research.cvbp@ggzdrenthe.nl; 2Department of Child and Adolescent Psychiatry, University of Groningen, University Medical Center Groningen, Hanzeplein 1, 9713 GZ Groningen, The Netherlands; p.hoekstra@accare.nl (P.J.H.); a.de.bildt@accare.nl (A.d.B.); 3Centre for Consultation and Expertise, Australielaan 14, 3526 AB Utrecht, The Netherlands; martinejansen@cce.nl; 4Accare Child Study Center, Groningerstraat 352, 9402 LT Assen, The Netherlands

**Keywords:** intellectual disability, challenging behaviour, service providers, consultation and expertise, diagnostic, treatment and support needs, AAIDD model on human functioning

## Abstract

Service providers may experience difficulties in providing appropriate care to optimize the functioning of individuals with intellectual disability and challenging behaviour. External consultation to identify and address the unmet support needs underlying the behaviour may be beneficial. Applying the multidimensional American Association Intellectual and Developmental Disabilities (AAIDD) model may facilitate this approach. We aimed to describe the content and outcomes of consultation for individuals with intellectual disability and challenging behaviour referred to the Dutch Centre for Consultation and Expertise in relation to the AAIDD model. Interventions were based on the clients’ diagnostic, treatment, and support needs and were categorized according to the five dimensions of the AAIDD model. Outcomes of the consultations were assessed based on reports in the file and rated as ‘clear improvement’, ‘improvement’ or ‘no improvement or deterioration’. In two-thirds of the 104 studied files, consultees were satisfied with the improvement in functioning. Interventions targeted the difficulties of the service providers in supporting their clients and were most often applied within the Health and Context dimensions of the AAIDD model. We may conclude that consultation of an expert team may be valuable to support the care providers, and the use of the AAIDD model may be helpful to address the unmet needs to improve the functioning of individuals with challenging behaviour.

## 1. Introduction

Challenging behaviour is defined as ‘behaviour of such an intensity, frequency or duration that the physical safety of the person or others is likely to be placed in serious jeopardy, or behaviour which is likely to seriously limit use of, or result in the person being denied access to, ordinary community facilities’ [1]. Many individuals with an intellectual disability present with challenging behaviours. Estimates vary from 18% in community populations [2] to 85% in people with profound intellectual and multiple disabilities living in institutions [3]. Challenging behaviours negatively affect an individual’s quality of life [4], social participation [5] and functioning in daily life [6]. Challenging behaviours may also cause a high burden and reduced quality of life among relatives [7,8], peers [9] and professional caregivers [10].

Both externalising behaviours (verbal and physical aggression, aggressive–destructive behaviours, inappropriate (sexual) behaviours, hyperactivity and/or irritability or other disruptive behaviours) and internalising behaviours (withdrawn behaviour, self-injurious behaviour, stereotypic behaviour and/or lethargy) may constitute challenging behaviours [11,12]. Individuals may present with multiple types of challenging behaviour at the same time and over time. The intensity and frequency may vary over time, but challenging behaviours are often present at a young age and are long lasting and persistent [12,13,14,15].

The presence and persistence of challenging behaviours in people with intellectual disability is associated with many factors, such as male sex, a higher degree of intellectual disability and higher care need [16,17], communication, sleep, sensory and motor problems, the presence of autism spectrum disorder and other (sometimes undiagnosed) mental and physical disorder [3,18,19,20]. In addition, environmental factors such as residential setting (both institutional and community living facilities versus living with families) and staff having difficulties in the management of challenging behaviours are related to the presence of challenging behaviours [16,21].

Given the variety of factors that may lead to challenging behaviours, interventions to reduce these behaviours should be applied from a multidisciplinary perspective, aiming to optimize the individual’s quality of life, including physical and mental functioning, emotional well-being and social participation [22]. An intervention should fit the individual’s specific needs, behaviours and situation and take the presumable causal and maintaining factors into account in order to understand at what point and in what way to intervene. However, these causal and maintaining factors can be difficult to identify since they differ among individuals and are often intertwined. In analysing challenging behaviours of individuals with intellectual disability and establishing the focus of interventions and support, it is, therefore, necessary to consider all factors that may be important. Because of the complexity of serious and/or long-standing challenging behaviours, the application of a model will facilitate unravelling all potential causal and maintaining factors and targeting interventions.

The American Association of Intellectual and Developmental Disabilities (AAIDD) [23] developed a multidimensional model to understand the functioning of people with intellectual disability based on the individual characteristics belonging to the dimensions of Intellectual functioning, Adaptive behaviour, Participation, Health and Context. The model is a functional model based on the International Classification of Functioning, Disability and Health model (ICF-model) [24]. It shows how the appropriate support (e.g., to compensate for limitations in Adaptive behaviour) may improve functioning (see Figure 1). The dimensions Health and Context are particularly important regarding the causes for challenging behaviours, as health issues and contexts that are not adjusted to the needs of an individual often underly (the persistence of) challenging behaviours. In turn, challenging behaviours may negatively influence the dimension Participation and the expression of practical adaptive and social skills from the dimensions Intellectual functioning and Adaptive behaviour. Therefore, the use of this model may be helpful in identifying the unmet support or treatment needs as leads for interventions to improve Human functioning outcomes.

To use the model among people with challenging behaviours, we operationalised this model by distinguishing three factors of the biopsychosocial model (i.e., physical, psychological and emotional health) [25,26,27] within the Health dimension and the three system layers of the ecological model (i.e., the macro-, meso- and microsystem) [28] within the Context dimension.

**Figure 1 ijerph-20-06365-f001:**
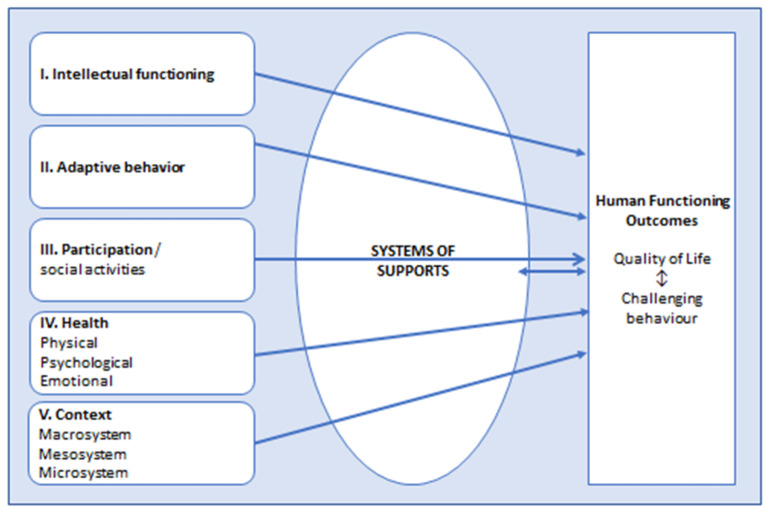
American Association of Intellectual and Developmental Disabilities model (AAIDD model, 2021) applied in people with intellectual disabilities who present with challenging behaviours. The dimensions Health and Context are operationalized by adding factors of the biopsychosocial model (Engel, 1972) [25] and systems of the ecological model (Bronfenbrenner, 1979) [28], respectively.

Regarding Context, Olivier-Pijpers et al. [29] found associations of challenging behaviours with factors at all three system layers, e.g., staff–member interactions with clients, staff’s sense of safety and support, staff turnover and understaffing and disability policies [29].

Especially for individuals with serious and/or persistent challenging behaviours and complex problems, it is important to thoroughly assess their functioning on all dimensions in the AAIDD model and to align these with the systems of support and outcomes as described above [23]. However, this is often easier said than done. These individuals may overchallenge the possibilities for assessment and the intervention of service providers and care professionals, leading the latter to experience difficulties in providing effective care [20,30,31]. Furthermore, even when evidence-based knowledge of the causes and treatment of challenging behaviour is available in the guidelines [22], its implementation in daily practice may be difficult. Wood et al. [32] found differences within and between care teams regarding the adherence to clinical standards. They also found that contextual factors, such as organizational culture and attitudes, may hinder the change in practice that is needed to implement evidence-based knowledge [32]. In some countries, external expertise is available, e.g., support or help from specialised intellectual disability healthcare teams, when internal analysis and/or intervention for challenging behaviours have not succeeded [27,33]. The organisations of healthcare services for individuals with intellectual disability varies across countries, and the effectiveness of the different organisations of services and interventions is not clear [34].

In The Netherlands, the Centre for Consultation and Expertise (CCE) has been designated by the Ministry of Health, Welfare and Sports to support service providers in providing care for individuals with serious challenging behaviours and complex needs. The CCE may be consulted by healthcare professionals, parents and families who experience difficulties in the assessment, treatment, management or guidance of the challenging behaviours of the individual for whom they care. The care responsibility always remains with the original care provider. The clients of the CCE are individuals with serious challenging behaviours in long-term care, including specialised youth, the elderly, and those under disability and mental healthcare. The CCE performs 1700 consultations per year, of which approximately 650 are for clients with an intellectual disability. After an application for consultation is approved, one of the 60 CCE coordinators is designated to the case. As the core member, the coordinator compiles a ‘consultation team’ from a pool of more than 600 experts/consultants. Usually, a behaviour scientist (psychologist or educational psychologist) is included; many times, a specialist nurse, an intellectual disability physician or a psychiatrist, and sometimes other experts such as a psychotherapist or a physio/sensorimotor therapist are also included. In essence, a consultation team is specifically tailored to the needs of the case. This may be a benefit, but it also has the risk of randomness since consultants may differ in expertise and adherence to professional standards. Nevertheless, in all their consultations, the CCE’s consultants take an integrative and multidisciplinary approach. The starting point is the inability of the clients’ care team to rise to the client’s behaviours that challenge and to find ways to support the team and care providers to find solutions to manage these behaviours and to optimize the clients’ functioning and quality of life. The consultation team considers all factors that may cause or maintain the challenging behaviours, ranging from client factors to contextual factors (a broad view). The client factors include the person’s own, unique story (an idiographic view). The consultation team not only considers facts but also includes the different perspectives (opinions, experiences and interests) of all persons and bodies/authorities in the client’s social environment (a layered view). Based on the relevant factors from the status, history and context of the client, the team drafts a working hypothesis (an idiographic theory) as the basis for finding leads for the interventions. A consultation often includes a multidimensional method of history taking and observation as well as interviewing of the client in his/her own environment (i.e., observation of the clients’ living situation and daily activities and interviews of the caregiver and family). The available files are studied to investigate whether additional diagnostics are necessary, e.g., a physical examination to complete all personal and contextual information for a broad and layered assessment of the client’s challenging behaviour.

Based on the analysis, the CCE makes recommendations on the diagnostic interventions as well as on the treatment/support interventions needed to reduce the challenging behaviour and improve the functioning. The CCE team members may coach the client’s own care team in performing and monitoring the treatment interventions, but they do not perform them directly. Consultants may perform diagnostic interventions themselves when the client’s own care professionals do not have the expertise. This approach may be beneficial in the sense of care teams learning to practise new knowledge. However, there may be difficulties in the implementation of the CCE’s recommendations at the team and/or organisation level as described by Wood et al. [32], leading to a postponement or cancellation of the diagnostic and treatment interventions. Indeed, a consultation and coaching process of the CCE may take several years to monitor the implementation.

The CCE has its own specific approach in assessments of challenging behaviour. This approach and their working methods may fit with the AAIDD model because the CCE adds their expertise to the care as delivered by the service providers by addressing the support needs of the individuals with challenging behaviour. Unfortunately, the CCE has no data on the effectiveness of their consultations, and the outcomes are yet unclear. It is also unclear in which specific dimensions the service providers experience difficulties when addressing the support needs of these clients [23]. In the current file study, we described the content and outcomes of the consultations in relation to the AAIDD model of adults and children with intellectual disability who consulted the CCE. We examined the client characteristics and how these related to the consultation characteristics according to the dimensions of the AAIDD model and to the outcomes. The focus was on the diagnostic and treatment interventions that the clients received and the extent to which these interventions were targeted to the clients’ needs on the various dimensions of the AAIDD model. The aim of this study was to explore the outcomes and content of the CCE consultations in order to increase the insight into the specific dimensions of the support needs of the clients that the care providers experience as difficult to meet and to provide recommendations to the service providers for improvements in identifying and addressing the support and treatment needs of clients with intellectual disability and challenging behaviours.

## 2. Materials and Methods

### 2.1. Materials

We aimed to study the files of all 274 individuals with a moderate, severe or profound intellectual disability who had been accepted for consultation to the CCE in 2016. Individuals with a moderate, severe or profound intellectual disability and challenging behaviour had a developmental age below 6 years, commonly combined with complex needs. We chose to include consultations that started in 2016, since these were most likely to be finished by the time data collection for this study was performed (in 2021). All eligible files were identified by an administrative CCE employee, and legal representatives of those 274 individuals were sent information letters about the study and asked to consent to the use of the files.

### 2.2. Ethics and Procedure

Because this was a file study, there was no need for IRB approval according to the Dutch legislations and regulations. Of course, we had to adhere to the Dutch and European laws concerning data protection rules. Therefore, informed consent of legal representatives regarding use of the clients’ file for this study was necessary.

We sent letters by mail to the legal representatives of the 274 eligible individuals, providing information about the study. The representatives were informed that data collection would take place by researchers who would adhere to the Dutch and European laws concerning general data protection rules, and that data would be encrypted and analysed anonymously. When the legal representative consented that the CCE file of the person they represented could be used in the study, they were requested to sign and send back the informed consent form.

### 2.3. Data Collection

We collected CCE file data on the following characteristics: demographics (age and sex), degree of intellectual disability (AAIDD dimensions I and II), daytime activities (dimension III), somatic and mental comorbidities, including chronic conditions and the types of challenging behaviours (dimension IV) and the living situation (institutional or at home with families; dimension V). These data were directly provided to the CCE consultants by clients’ healthcare professionals (e.g., nurses, physicians and psychologists or educational psychologists). Commonly, the degree of intellectual disability was based on psychological examination by registered psychologists commissioned by the Centre Indication Care (in Dutch CIZ). Characteristics of the CCE consultations were retrieved from the files, i.e., administrative data and notifications from free text fields and standardized progression and advisory reports of the CCE coordinator and consultants. These included the reason for referral as categorized by the CCE, whether the client had consulted the CCE before (i.e., re-referral), the number of the involved consultants and their professions, the diagnostic interventions the consultants recommended or performed and the treatment interventions they recommended, the duration of the trajectory and the outcome of the consultation. All interventions were monitored by the CCE coordinator and guided or coached by the CCE consultants.

The outcomes of the consultation were assessed by the second author in consultation with the first author and were based on the content of progression and advisory reports within the CCE file, which were discussed with the clients’ representatives and professional caregivers and on the CCE coordinators’ notifications in the file. We rated the outcomes as (1) clear improvement (a clear improvement in the behaviour of the client and/or functioning at the end of the CCE consultation, all recommended interventions were implemented and consultees’ questions were satisfactorily addressed); (2) improvement (although additional interventions were needed which could, however, be applied without further involvement of the CCE, and/or not all questions were addressed); and (3) no improvement or deterioration in behaviour and/or functioning and/or recommended interventions were not or insufficiently implemented and/or consultees’ questions were unsatisfactorily addressed.

The diagnostic and treatment/support interventions were categorized by the second author in consultation with the first author according to the operationalized dimensions of the AAIDD model (Figure 1), e.g., the diagnostic assessment ‘physical examination’ was categorized under the dimension Health/Physical, and the intervention ‘education and training of staff’ (which supports the fitness of clients’ context) was categorized under the dimension Context/microsystem. We categorized diagnostics and interventions that aimed to improve the communication and social–emotional functioning of a client within the dimension Adaptive behaviour.

### 2.4. Statistical Analyses

We computed the frequencies of categorical variables and means of continuous variables for the client (demographic, health and context variables) and consultation characteristics (reason for referral, type and number of consultants, interventions and duration and outcome of consultations). We investigated possible differences in client and consultation characteristics between adults and children/youth with Pearson Chi-square tests and t-tests, respectively. To investigate correlations between client and consultation characteristics, Pearson correlation tests were used for continuous variables.

Univariate linear regression analyses were used to investigate associations between consultation characteristics and outcomes of the consultation trajectories. Variables associated at the *p* < 0.1 level were included in a multivariate linear regression analysis to investigate the multifactorial model. All analyses were performed in IBM SPPS Statistics 26. *p*-values < 0.05 were regarded as statistically significant.

## 3. Results

### 3.1. CCE Files

We received informed consent from legal representatives to study 104 of the potential 274 eligible CCE records. Of those, 72 (69%) were adults, and 32 (31%) were children/youth.

### 3.2. Client and Consultation Characteristics

Table 1 displays the clients’ characteristics and consultation characteristics of adults and children/youth. There were significant differences between these two groups. Adults were less likely to be male and live at home, and they had more (and other comorbid) mental conditions than children. In particular, affective, anxiety, psychotic and sensory integration disorders were more often present in adults. In addition, adults had a higher incidence of chronic dermatological conditions and mental conditions than children. Furthermore, the consultants’ profession ‘social worker’ was more prevalent in children/youth than adults (16% and 3%, respectively, *p* = 0.02).

The mean number of consultants in the 104 consultations was 2.6 (SD 1.1, range 1–7), including the CCE coordinator/behavioural scientist. In 53% of cases, a psychologist was involved; in 14%, a psychiatrist; and in 11%, an intellectual disability physician. Other involved disciplines were case manager (14%), physio/sensorimotor therapist (14%), nurse (10%), social worker (7%), physiotherapist (6%), psychomotor therapist (7%) and systemic/family therapist (7%). More than one-third of the consultations were re-referrals, in children as well as in adults. In the consultations, interviews and history taking were conducted in 75% of the cases, observation and/or video of the clients in their context in 73% and the request for and study of additional files in 66%.

Outcomes in two-thirds of the cases were assessed from the files as favourable, i.e., the clients’ functioning had clearly improved and/or the consultees’ questions were satisfactorily addressed by the CCE. In approximately one-quarter, the results were likely favourable, i.e., there was an improvement, but some remaining interventions were needed in which service providers indicated they could implement these recommendations by themselves without the CCE’s support and/or not all questions were satisfactorily addressed, and in approximately 10%, outcomes were less favourable, i.e., there was no improvement and/or the consultees’ questions were unsatisfactorily addressed.

Table 2 shows the diagnostic and treatment/support interventions which were performed by the CCE consultants themselves and/or by clients’ own healthcare professionals as recommended by consultants during the consultations, categorized according to the five dimensions of the AAIDD model.

Most of the recommended interventions were applied within the dimensions Health and Context. Regarding Health, the treatment of physical disorders included paramedic therapy. Change in prescriptions of medication included change in psychotropics (28%), anti-epileptics (11%) and other medication (10%). Regarding the dimension Context, changes in the microsystem were most often the explanation of the needs of the clients and the training of the caregivers to adapt their guidance and communication style to the needs of their client. Recommended diagnostic health interventions were often re-assessing, expanding or updating the diagnostics of chronic conditions such as pain, epilepsy, sleep and mood disorders; diagnostic context interventions were often the analysis of support needs and systems, communication, staff/client and staff/client’s systems interactions, team collaboration and team support. Recommended treatment health interventions were often medication changes, applying sensorimotor or other body-oriented therapies or adjustments of daily activities. Recommended treatment context interventions were often education of teams regarding the specific symptoms and treatments of psychopathology and/or challenging behaviour of the clients and the training of communication and treatment skills in responding to symptoms and behaviours. Often, multiple diagnostic or treatment/support interventions were applied for one case, sometimes covering various dimensions. Furthermore, diagnostic interventions within one dimension could result in treatment/support interventions within another dimension. For example, diagnosing deafness (dimension Health/Physical) resulted in the application of communication aids and training of the client in using these aids, e.g., use of icons (dimension Adaptive behaviour).

Some of the interventions were more often applied in adults than in children/youth, i.e., psychotropic drug prescription in 35% compared with 12.5% in children/youth (Pearson Chi-square = 5.44; *p* = 0.02); education of system in 68% compared with 47% (Pearson Chi-square = 4.20; *p* = 0.04; and adapting the physical environment in 40% of the adults compared with 16% in children/youth (Pearson Chi-square = 6.12; *p* = 0.01).

### 3.3. Relationships between Client and Consultation Characteristics

Because we aimed to provide recommendations to the service providers in identifying and addressing the support and treatment needs of the clients with intellectual disability and challenging behaviours, we also investigated the potential relationships between client and consultation characteristics. When comparing groups based on the reason for consultation (as noted by the CCE, described in Table 1), there were no differences in client characteristics. Some client characteristics were related to the consultants’ professions. A nurse was more often involved when a urogenital disorder was present (71% versus 29%, Pearson Chi-square = 6.69, *p* = 0.01), a social worker more often in the presence of sensory–integration problems (16% versus 4%, Pearson Chi-square = 4.50, *p* = 0.03), a psychiatrist in the case of affective disorders (26% versus 10%, Pearson Chi-square = 4.50, *p* = 0.048) and a system therapist and a psychomotor therapist in the case of destructive behaviour (16% versus 3%, Pearson Chi-square = 6.25, *p* = 0.01; 19% versus 1%, Pearson Chi-square = 10.21, *p* = 0.001, respectively).

We found no correlations of age or number of somatic, mental and behavioural conditions with the duration of the consultation trajectories nor with the number of consultants.

### 3.4. Relationships between Characteristics of the Consultations and Outcomes

Because we sought to explore the outcomes of the CCE consultations, we investigated the potential relationships between consultation characteristics and outcomes. Involvement of the CCE consultant profession ‘intellectual disability physician’ was negatively associated and that of the profession ‘psychiatrist’ was positively associated at the *p* < 0.1 level with a better outcome of the consultation, i.e., the reasons for referral were satisfactorily addressed, and/or clients showed more improvement in behavioural and/or overall functioning. The advice to stop or to start medication other than psychotropics and/or anti-epileptics and/or to lower or raise the dosage of this medication was negatively associated at the *p* < 0.1 level with the outcomes of the consultations. The interventions ‘the writing of an alert, crisis and action plan’ and ‘change in caregivers’ guidance and communication style’ were positively associated at the *p* < 0.1 level with the outcomes of the consultations.

These variables were included in a multivariate analysis to investigate a multifactorial model. This analysis showed that the involvement of a psychiatrist as a CCE consultant (β = 0.20, t = 2.12, *p* = 0.04) and the deployment of education and training to adapt the caregivers’ guidance and communication style to their client’s needs (β = 0.22, t = 2.32, *p* = 0.04) were associated with better outcomes. Additionally, the advice to stop or to start medication other than psychotropics and/or anti-epileptics and/or to lower or raise the dosage of this medication was associated with worse outcomes (β= −0.23, t= −2.28, *p* = 0.03). That is, the consultees’ questions were not (or not completely) addressed, and/or there was less or no improvement or deterioration in behavioural and/or overall functioning.

## 4. Discussion

Presented here was a file study of 104 clients with intellectual disability and complex needs who had been referred by professionals affiliated with intellectual disability service providers to an expert consultation centre (CCE) for improving the functioning of individuals with persistent challenging behaviours. The reasons for consultation were mainly diagnostic and treatment questions to address the clients’ needs for support and to seek advice for appropriate guidance and living environment. We investigated the consultation questions, content and outcomes in relation to the client characteristics. We used the internationally well-known AAIDD functioning model to categorize the dimensions on which the CCE’s expertise in diagnostics and treatment was targeted as an indication of which clients’ needs were (not) met in their own facility. In addition, the CCE’s interventions of supporting the client’s own caregivers in addressing these unmet needs to improve the clients’ (behavioural) functioning were categorized according to this model.

The first important finding of our study shows that the involvement of an expert consultants team for individuals with intellectual disability, challenging behaviours and complex needs seems valuable. We found that consultees, i.e., representatives and professional caregivers of clients, were satisfied with the outcomes of the consultation in approximately two-thirds of the cases, as assessed from the CCE reports and notifications in the file, meaning that the client’s challenging behaviours had decreased and/or their functioning had improved and/or that all consultees’ questions were addressed and clarified. Apparently, external consultation is helpful in improving the identification of all unmet needs related to the client’s challenging behaviour. In almost one-quarter of the consultations, there was improvement, but more interventions were needed to address the clients’ unmet needs, which could be implemented without the support of the CCE, and/or there were some questions that were not satisfactorily addressed.

Additionally, our study showed that the CCE approach was in line with the AAIDD model in that consultants had taken a multidimensional and integrative view in their assessments, and the interventions were directed to reducing the challenging behaviour and improving the functioning. The finding that actual diagnostic and treatment interventions or recommendations of the consultants did not cover all dimensions in all cases can be explained by the fact that the consultation team aims to provide case-tailored support, using all diagnostic information that is already available and coordinating the CCE interventions with the care that the clients already receive from their own care professionals. This may explain why the Intellectual functioning and Adaptive behaviour dimensions were not often recommended in diagnostic interventions (28% and 15%, respectively), likely because this information was already available in the provided files.

In addition, the model revealed the specific dimensions where the service providers needed the CCE’s expertise in identifying the client’s unmet needs and to intervene to address the client’s needs for support and/or treatment for optimal functioning. For example, in the present study, more diagnostics were needed in the clients’ healthcare, education of staff in the clients’ context and changes in day-care activities for better social participation. The most frequent diagnostic interventions that were performed or recommended fell within the Health/Physical and Health/Psychological dimensions. This is not surprising based on the high percentages of somatic and mental comorbidities in the sample (80–90%). These comorbidities may interfere with or underly the challenging behaviours. The presence of physical and mental health problems is known to be related to aggressive behaviour [3,11,18], which was the most frequent type of challenging behaviour, and to self-injurious behaviour (the second-most frequent type). However, it may be surprising that even though in Dutch intellectual disability healthcare, there are specialized intellectual disability physicians and experienced behavioural scientists available, the clients’ needs for treatment or support on this dimension are not always met, resulting in their challenging behaviour. Moreover, staff and caregivers may not be aware of the implications of, or need support to implement, the diagnostic information on the guiding needs and daily functioning of their clients. This may be illustrated by the finding that the most frequent treatment interventions fell within the Context and Participation/Social activities dimensions, e.g., education and training of caregivers, coaching of teams in adapting their guidance and communication style to the specific needs of their clients with challenging behaviours and adapting the day-care activities especially of those clients who live with their relatives. Although system problems were a reason for referral in only approximately 5% of the cases, system interventions were recommended or applied in 33% of the cases.

Finally, we found associations between consultation characteristics and the outcomes of the consultations. First, when the intervention was directed at the microsystem to address the individual’s needs and adapt their support and communication style, the outcome was better. Second, outcomes also were better when a consultant psychiatrist was involved. As described by Axmon et al. [35], a psychiatric diagnosis is less likely to be diagnosed in the case of challenging behaviours in more severe intellectual disability [35], and our findings indicate that involving a psychiatrist may lead to a better understanding of the possible underlying mental disorders and, therefore, a better fit of guiding recommendations. The finding that changes in medication other than anti-epileptics and psychotropics is associated with worse outcomes may be related to the identification of progressive physical disorders, e.g., syndromic metabolic or neurodegenerative disorders, which are relatively often present in our study population. These disorders are often associated with challenging behaviours and progressive impaired functioning despite pharmacological treatments. However, we have no data to substantiate this explanation.

As far as we know, our study is the first to explore the use of the AAIDD model in the care for individuals with intellectual disability. Alford et al. [24] concluded in a systematic review on the implementation of the ICF model (on which the AAIDD model is based) that this model is a useful tool to identify and address people’s unmet needs, which may improve health care. The results of our study may indicate that the AAIDD model is also a useful tool in healthcare for people with intellectual disability.

### Limitations and Strengths

In this retrospective file study, we met some limitations which should be considered when interpreting the results of this study. First, the standardised and validated outcome measures of the effect of interventions are missing, and outcomes had to be broadly assessed from information in the files. Second, we had no data on the criteria which were used by the CCE coordinator in the formation of the multidisciplinary consultant team and the decision to close the consultation. In addition, data were lacking regarding the professional standards that were used by the consultants in their diagnostic and treatment/support interventions and how the effects of these were monitored. Additionally, although the CCE consultations may last for several years, there may be risks of shortcomings in sustainably anchoring the implementation of the contextual interventions, such as changes in the working methods, staff education and coaching in the management of challenging behaviours due to the contextual problems of the service providers [31,32]. Furthermore, the fact that we found no relationship between the reasons for consultation and the content and outcomes of consultations is likely due to the categories for reasons of consultation (as assigned by the CCE) that highly overlap in content. Last, we received informed consent for 104 out of the 274 eligible records, and we do not know whether there were differences between client and consultation characteristics from these files and the 160 files not included in the study. Therefore, the results of this study may not represent all CCE consultations regarding this group of clients.

A strength of our study was the value for clinical practice and healthcare policies in the care for individuals with intellectual disability and challenging behaviour. The results represent the clinical practice of caregivers and healthcare professionals having concerns regarding the quality of life and (behavioural) functioning of the person for whom they care and their struggle regarding how to improve these. Our study clearly shows that resources should be sought for an improvement in appropriate healthcare and contextual support.

## 5. Conclusions

This study shows that involvement of an expert consultant team with individuals with intellectual disability, challenging behaviours and complex needs may be valuable. In two-thirds of the 104 cases, consultants noted that the consultees’ questions were addressed, the individual’s functioning had improved and/or the client’s context had become more adjusted to their needs.

The use of the AAIDD model in exploring the content of consultations in our study showed that in assessments, consultants had taken a multidimensional and integrative view. In addition, the model revealed the specific dimensions where service providers had to intervene to address the client’s needs for support and/or treatment for optimal functioning. In particular, contextual and health interventions were often applied. Results suggest that the education and support of staff regarding the communication and guidance of the client are associated with reducing the challenging behaviours and improving the clients’ functioning, even though these are almost never the direct reasons for consultation.

We strongly recommend that service providers invest in care teams for clients with challenging behaviours, not just regarding education and training in diagnosis and treatments of (psycho)pathology, behavioural therapies and guidance of these clients but also in team coaching and support. Furthermore, when service providers know which specific professional expertise should be deployed for identifying and meeting these clients’ support needs, they can adapt their personnel policies to provide effective care. However, more studies are needed to investigate which client and contextual characteristics are related to the care pathways these clients should be offered. Building upon our current findings, the next step would be to perform prospective studies with robust methodologies on the effectiveness of the consultants’ interventions and seen in the light of client’s functioning by using the AAIDD model.

## Figures and Tables

**Table 1 ijerph-20-06365-t001:** Characteristics of clients with a moderate, severe or profound intellectual disability who had been referred to an expert centre because of persistent challenging behaviours.

	Children/Youth <18 Years *n* = 32	Adults ≥ 18 Years *n* = 72	Significant Difference ^#^
Age in years, mean (SD)	11.9 (3.9)	38.7 (14.1)	not applicable
Sex (male)	78%	55%	*p* = 0.028
Degree of intellectual disability ^&^:			
Moderate	42%	35%	
Severe	39%	34%	
Profound	19%	28%	
Developmental age below 6 years		3%	
Living situation:			*p* = 0.001
With relatives	75%	11%	
Community care centre	3%	22%	
Congregated care centre/institution	22%	67%	
Presence of daytime activities	100%	92%	
Presence of somatic problems	84%	90%	
Hearing	16%	21%	
Visual	28%	35%	
Epilepsy	38%	27%	
Cerebral palsy	25%	22%	
Genetic	38%	28%	
Gastro-intestinal	41%	32%	
Urogenital	6%	17%	
Respiratory	3%	4%	
Dermatological	0%	17%	*p* = 0.01
Number of somatic conditions, mean (SD)	2.6 (1.6)	2.8 (1.9)	
Presence of mental comorbidity	78%	92%	
Affective disorders	12.5%	32%	*p* = 0.04
Anxiety disorders	3%	35%	*p* = 0.001
Sensory integration disorders	15%	44%	*p* = 0.002
Sleep disorders	34%	42%	
Psychoses/delusions	0%	14%	*p* = 0.03
Autism spectrum disorder	50%	53%	
Attention Deficit Hyperactivity Disorder	9%	4%	
Number of mental conditions, mean (SD)	1.7 (1.2)	2.4 (1.4)	*p* = 0.02
Presence of challenging behaviours:	91%	99%	
Aggressive behaviour	63%	78%	
Verbal aggression	44%	47%	
Destructive behaviour	31%	29%	
Defiant/oppositional behaviour	28%	18%	
Hyperactivity	19%	14%	
Lethargy	6%	11%	
Self-injurious behaviour	47%	46%	
Number of challenging behaviours, mean (SD)	3 (1.9)	3 (1.3)	
Reason for consultation:			
Diagnostics and treatment advice	60%	47%	
System problems/social context	6%	4%	
Advice for appropriate care provision	34%	40%	
Diagnostics and treatment to improve the physical and mental functioning	0%	9%	
Re-referral	38%	36%	
Number of consultants, mean (SD)	2.7 (1.4)	2.5 (0.9)	
Duration of consultation in months,			
mean (SD)	41.8 (25.6)	38.8(20.4)	
Outcome of consultation, mean (SD) *	2.6 (0.7)	2.6 (0.6)	
Improvement/consultees’ questions satisfactorily addressed	66%	65%	
Some improvement/not all questions addressed	25%	21%	
No improvement/questions unsatisfactorily addressed	9%	7%	

^#^ Chi-square test; ^&^ based on psychological examination; * Adults: missing 7%.

**Table 2 ijerph-20-06365-t002:** Diagnostic and treatment interventions, categorized according to the dimensions of the model of the American Association of Intellectual and Developmental Disabilities (AAIDD model; Schalock, 2021) [23], during consultations of an expert centre in clients (*n* = 104) with moderate, severe or profound intellectual disability and persistent challenging behaviours.

		Diagnostic Interventions Total % in Whole Sample	Treatment Interventions Total % in Whole Sample	Diagnostic and Treatment Interventions within AAIDD Dimension by Case ^3^ Total % in Whole Sample
**Dimension: Intellectual functioning**				10%
Psychological examination of intellectual abilities		28%		
**Dimension: Adaptive behaviour**				39%
Examination of social–emotional development		14.5%		
Applying communication aids			18%	
Training ADL skills			11%	
**Dimension: Participation/social activities**				44%
Adaptation/change in day-care activities			44%	
**Dimension: Health ^1^**		Diagnostic 42%	Treatment 62%	
Physical				57%
*Genetic testing*		15%		
*Physical examination*		35%		
*Medication (psychotropics and other)*			49%	
*Physical disorders*			11%	
*Applying medical devices or tools*			2%	
Psychological			18.5%	40%
*Sleep disorders*		37.5%		
*Psychopathology*		15%		
*Individual psychological therapy*		22%	15%	
*Advice regarding sleep hygiene*			3%	
Emotional			26%	23%
*Applying of alert, crisis and action plan*			14%	
*Adjustment of daily activities*			12%	
**Dimension: Context ^2^**				
Macrosystem				11%
*Extra financial and/or personal resources*			11%	
Mesosystem				25%
*Adaptation of physical environment/living situation*			33%	
Microsystem				89%
*Team coaching/education*			63%	
*Team training to adapt guidance and communication*			64%	
*System therapy*			33%	

^1^ Differentiated in factors of the biopsychosocial model (Engel, 1977). ^2^ Differentiated in system layers of the ecological model (Bronfenbrenner, 1977). ^3^ One case may include multiple interventions spread over one or multiple dimensions. ADL—Activities of Daily Life.

## Data Availability

The data are available on request and with a substantial explanation at GGZ Drenthe/department research https://ggzdrenthe.nl/research (accessed on 14 December 2022).

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
