# Peer review of "Difficulties in Addressing Diagnostic, Treatment and Support Needs in Individuals with Intellectual Disability and Persistent Challenging Behaviours: A Descriptive File Study of Referrals to an Expertise Centre"

_ijerph, 2023, doi:10.3390/ijerph20146365_

Round 1

Reviewer 1 Report

Dear Authors,

According to the title, the manuscript was devoted to up-to-date problem, i.e. providing an appropriate care for mentally retarded people. But authors did not present any information about the scientific aim of the study. In the section Materials, I found extremely scarce information (clinical and psychological) about the participants. Any qualitative information about the multiple mental health measures which would be statistically tested was not presentedmethods of the assessment were not discribed. In total, the manuscript does not contain any new scientific information.

Author Response

Dear reviewer 1,

We thank the reviewer for the willingness to review our manuscript and the comments. Below you will find our responses. 

Yours sincerely, 

Gerda de Kuijper (on behalf of the co-authors)

Reviewer 1

1) According to the title, the manuscript was devoted to up-to-date problem, i.e. providing an appropriate care for mentally retarded people. But authors did not present any information about the scientific aim of the study.

We apologize for this and agree with the reviewer about the missing of the aim of the study. We added a sentence about the topic of our study to the Introduction section and added our aim at the end of the Introduction section:

Research topic:

The CCE has its own specific approach in assessments of challenging behaviour. This approach and their working methods may fit with the AAIDD-model because the CCE adds their expertise on addressing support needs of individuals with challenging behaviour to the care as delivered by service providers. ‘Unfortunately, the CCE has no data on the effectiveness of their consultations and the outcomes are yet unclear. It is also unclear in what specific dimensions service providers experience difficulties when addressing support needs of these clients.’

Research aim:

‘ The aim of this study was to explore the outcomes and content of the CCE-consultations, in order to increase the insight into the specific dimensions of support needs of clients that care providers experience as difficult to meet, and to provide recommendations to service providers for improvements in identifying and addressing support and treatment needs of clients with intellectual disability and challenging behaviours.

We also slightly changed the title to clearer indicate that we studied the difficulties of care providers in providing appropriate care to their clients with challenging behaviour. The new title is:

‘Difficulties in addressing diagnostic, treatment and support needs of individuals with intellectual disability and persistent challenging behaviour: a descriptive file study of referrals to an expertise centre ’.

 2) In the section Materials, I found extremely scarce information (clinical and psychological) about the participants. Any qualitative information about the multiple mental health measures which would be statistically tested was not presented; methods of the assessment were not described. In total, the manuscript does not contain any new scientific information.

Table 1 provides characteristics of the individuals with intellectual disability whose files we investigated.

We agree with the reviewer that we do not have a validated outcome measure and have added this as a limitation to the discussion section:

‘First, standardised and validated outcome measures on the effect of interventions are missing and outcomes had to be broadly assessed from information in the files.’

In the revised manuscript we now better describe the methods of the assessment of the outcome measures by adding information about the extent to which the CCE’s recommended interventions were implemented :

‘ We rated the outcomes as 1) clear improvement (a clear improvement in the behaviour of the client and/or functioning at the end of the CCE consultation, all recommended interventions were implemented and consultees’ questions were satisfactorily addressed; 2)  improvement, but additional interventions were needed which could, however, be applied without further involvement of the CCE, and/or not all questions were addressed); and 3) no improvement or deterioration in behaviour and/or functioning, and/or recommended interventions were not or insufficiently implemented and/or consultees’ questions were unsatisfactorily addressed. ‘

We do not agree with the reviewer that our manuscript does not contain any new scientific information. On the contrary, since service providers often experience difficulties in addressing the support needs of clients with challenging behaviour, our study offers new knowledge on the specific AAIDD dimensions to which more attention should be paid  in addressing the support and treatment needs underlying these behaviours. We also provide recommendations to improve policies in addressing these support needs. See the last sentence of the discussion section:

‘Our study clearly shows that resources for improvement should be sought in better attention for appropriate healthcare and contextual support.’

 And the new phrase in the Conclusion section:

‘We strongly recommend service providers to invest in care teams of clients with challenging behaviours, not just regarding education and training in diagnosis and treatments of (psycho)pathology, behavioural therapies and guidance of these clients, but also in team coaching and –support. Furthermore, when service providers know which specific professional expertise should be deployed on identifying and meeting these clients’ support needs, they can adapt their personnel policies to provide effective care.’

Reviewer 2 Report

Dear authors, 

Below are the comments for each section of the article,

Introduction

The introduction provides a clear and concise overview of the topic, especially on discussing the challenging behaviours in individuals with intellectual disability and the authors have highlighted the importance of a multidisciplinary approach to address the complex needs of these individuals. However, there are several areas where the literature review could be improved.

-       The introduction section could benefit from a more comprehensive and critical examination of the literatures. The authors can further elaborate on the research gap, discussing the conflicting findings and unresolved questions on addressing the needs of individuals with intellectual disability. 

-       While the authors briefly describe the CCE's consultation process and how it relates to the AAIDD-model, they do not examine the effectiveness of this approach or explore any potential limitations or criticisms of the CCE's services. A more critical analysis of the CCE's approach would provide a more balanced and informative perspective on the topic.

-       The research questions and hypotheses didn’t mention explicitly. The aim of the current study is not specified as well. 

Materials and Methods 

-       Please further elaborate on the selection criteria for the study participants, especially how the individuals were identified to be included in the study.

-       It is suggested that the authors should specific variables that were included in the analyses or how they were analyzed later in the results sections.

Results and Discussion 

-       A more detailed description of the interventions would provide a clearer understanding of the types of treatments and support that were provided to the participants.

-       Apart from the analyses being done, a more in-depth analysis of the data would provide a more robust understanding of the findings, reporting the significance test results. Addition, explain how the analyses performed were related to the aim of the study. 

-       A more detailed discussion of the practical implications of the findings would be beneficial for healthcare professionals and policymakers, for example, any specific recommendation should be proposed. 

The writing is clear and concise. The article is well-written. 

Author Response

Dear reviewer 2,

We thank you for your willingness to review our manuscript and the comments. Below you will find our responses.

Yours sincerely,

Gerda de Kuijper (on behalf of the authors)

Reviewer 2

Introduction

The introduction provides a clear and concise overview of the topic, especially on discussing the challenging behaviours in individuals with intellectual disability and the authors have highlighted the importance of a multidisciplinary approach to address the complex needs of these individuals. However, there are several areas where the literature review could be improved.

1) The introduction section could benefit from a more comprehensive and critical examination of the literatures. The authors can further elaborate on the research gap, discussing the conflicting findings and unresolved questions on addressing the needs of individuals with intellectual disability.

We elaborated on the research gap by adding and discussing a literature reference about the difficulties in implementation of knowledge regarding addressing the needs of clients of intellectual disability service providers:

‘Furthermore, even when evidence-based knowledge on the causes and treatment of challenging behaviour is available in guidelines [22], implementation in daily practice may be difficult. Wood et al. [32] found differences in practice within and between care teams regarding adherence to clinical standards. They also found that contextual factors like organizational culture and attitudes may hinder the change in practice that is needed to implement evidence-based knowledge [32].

Reference 32:

‘32) Wood S., Gangadharan S., Tyrer F., Gumber R., Devapriam J. Hiremath A. & Bhaumik S. Successes and challenges in the implementation of care pathways in an intellectual disability service: health professionals’ experiences. J. Policy Pract Intellect Disabil., 2014, 11, 1-7.’

2) While the authors briefly describe the CCE’s consultation process and how it relates to the AAIDD-model, they do not examine the effectiveness of this approach or explore any potential limitations or criticisms of the CCE’s services. A more critical analysis of the CCE’s approach would provide a more balanced and informative perspective on the topic.

We thank the reviewer for this comment.

 We added some sentences/phrases:

‘In essence, a consultation team is specifically tailored to the needs of the case. This may be a benefit, but also has the risk of randomness since consultants may differ in expertise and adherence to professional standards.’

and:

‘Based on the analysis, the CCE recommends on diagnostic interventions as well as on treatment/support interventions needed to reduce the challenging behaviour and improve the functioning. The CCE team members may coach the client’s own care-team in performing and monitoring treatment interventions, but do not perform them directly. Consultants may perform diagnostic interventions themselves when the client’s own care professionals do not have the expertise. This approach may be beneficial in the sense of care teams learning to practise new knowledge. However, there may be difficulties in implementation of CCE’s recommendations at the team and/or organisation level as described by Wood et al.[32] leading to postponement or cancellation of diagnostic and treatment interventions. Indeed, a consultation and coaching process of the CCE may take several years to monitor the implementation.’

and:

The CCE has its own specific approach in assessments of challenging behaviour. This approach and their working methods may fit with the AAIDD-model because the CCE adds their expertise on addressing support needs of individuals with challenging behaviour to the care as delivered by service providers. Unfortunately, the CCE has no data on the effectiveness of their consultations and the outcomes are yet unclear. It is also unclear in what specific dimensions service providers experience difficulties when addressing support needs of these clients [23].

3)The research questions and hypotheses didn’t mention explicitly. The aim of the current study is not specified as well.

See our response to reviewer 1.

We apologize for this and agree with the reviewer about the missing of the aim of the study. We added a sentence about the topic of our study to the Introduction section and added our aim at the end of the Introduction section:

Research topic:

The CCE has its own specific approach in assessments of challenging behaviour. This approach and their working methods may fit with the AAIDD-model because the CCE adds their expertise on addressing support needs of individuals with challenging behaviour to the care as delivered by service providers. ‘Unfortunately, the CCE has no data on the effectiveness of their consultations and the outcomes are yet unclear. It is also unclear in what specific dimensions service providers experience difficulties when addressing support needs of these clients.’

Research aim:

‘ The aim of this study was to explore the outcomes and content of the CCE-consultations, in order to increase the insight into the specific dimensions of support needs of clients that care providers experience as difficult to meet, and to provide recommendations to service providers for improvements in identifying and addressing support and treatment needs of clients with intellectual disability and challenging behaviours.

We also slightly changed the title to clearer indicate that we studied the difficulties of care providers in providing appropriate care to their clients with challenging behaviour. The new title is:

‘Difficulties in addressing diagnostic, treatment and support needs of individuals with intellectual disability and persistent challenging behaviour: a descriptive file study of referrals to an expertise centre ’.

Materials and Methods

4) Please further elaborate on the selection criteria for the study participants, especially how the individuals were identified to be included in the study.

We did not use other selection criteria than those mentioned in the manuscript (files of those individuals who had been accepted for consultation to the CCE in 2016 with a moderate, severe or profound intellectual disability)

We added a sentence about the  identification of the eligible files that could be included in the study  at the end  of this section:

‘All eligible files were identified by an administrative CCE-employee and legal representatives of those 274 individuals were sent information letters about the study and asked to consent with the use of the files. ‘

5) It is suggested that the authors should specific variables that were included in the analyses or how they were analyzed later in the results sections.

We have clearly written this in our manuscript. In section 2.4 Statistical analyses we wrote:

“Univariate linear regression analyses were used to investigate associations be-tween the various consultation characteristics and outcomes of the consultation trajectories. Variables associated at the p<0.1 level were included in a multivariate linear regression analysis to investigate the multifactorial model.”

And in the section 3.4 we wrote:

“The interventions ‘the writing of an alert, crisis and action plan’ and ‘change in care-givers’ guiding and communication style’ were positively associated at the p<0.1 level with the outcomes of the consultations.

These variables were included in a multivariate analysis to investigate a multifactorial model.”

Moreover, we now specified the descriptive variables also in subsection 2.4 Statistical analyses:

‘We computed the frequencies of categorical variables and means of continuous variables for the client (demographic, health and context variables) and consultation characteristics (reason for referral, type and number of consultants, interventions and duration and outcome of consultations).’

Results and Discussion

6) A more detailed description of the interventions would provide a clearer understanding of the types of treatments and support that were provided to the participants.

We added this information and hope this will clarify the types of treatment and support the clients needed.

‘Most of the recommended interventions were applied within the dimensions Health and Context. Regarding Health, treatment of physical disorders included paramedic therapy. Change in prescriptions of medication included change in psychotropics (28%), anti-epileptics (11%) and other medication (10%). Regarding the dimension Context, changes in the microsystem were most often explanation of the needs of clients and training of caregivers to adapt their guiding and communication style to the needs of their client. Recommended diagnostic health interventions were often re-assessing, expanding or up-dating the diagnostics of chronic conditions like pain, epilepsy, sleep and mood disorders; diagnostic context interventions were often the analysis of support needs and systems, communication, staff/client and staff/client’s systems interactions, team collaboration and team support. Recommended treatment health interventions were often medication changes, applying sensorimotor or other body-oriented therapies, or adjustments of daily activities. Recommended treatment context interventions were often education of teams regarding the specific symptoms and treatments of psychopathology and/or challenging behaviour of clients, and training of communication and treatment skills in responding to symptoms and behaviours.’

7) Apart from the analyses being done, a more in-depth analysis of the data would provide a more robust understanding of the findings, reporting the significant tests results.

We added a sentence to the results section reporting the significant test results:

‘Especially affective, anxiety, psychotic and sensory integration disorders were more often present in adults. Also, adults had more often chronic dermatological conditions and mental conditions than children. Furthermore, (data not shown), the consultants’…

Addition, explain how the analyses performed were related to the aim of the study.

We added this information at the beginning of subsection 3.3. Relationships between client and consultation characteristics:

‘Because we aimed to provide recommendations to service providers in identifying and addressing support and treatment needs of clients with intellectual disability and challenging behaviours, we also investigated potential relationships between client and consultation characteristics.’

And at the beginning of subsection 3.4 Relationships between characteristics of the consultations and outcomes:

‘Because we wanted to explore the outcomes of CCE consultations, we investigated potential relationships between consultation characteristics and outcomes.’

8) A more detailed discussion of the practical implications of the findings would be beneficial to healthcare professionals and policymakers, for example, any specific recommendation should be proposed.

We thank the reviewer for this suggestion and added a phrase to the Conclusion section:

 ‘We strongly recommend service providers to invest in care teams of clients with challenging behaviours, not just regarding education and training in diagnosis and treatments of (psycho)pathology, behavioural therapies and guidance of these clients, but also in team coaching and –support. Furthermore, when service providers know which specific professional expertise should be deployed on identifying and meeting these clients’ support needs, they can adapt their personnel policies to provide effective care. However, more studies are needed to investigate which client and contextual characteristics are related to the care pathways  these clients should be offered.’

Reviewer 3 Report

I would have preferred an objective measure of statistical improvement

I prefer to put the word descriptive study in the title

Author Response

Dear reviewer 3,

We thank you for your willingness to review our manuscript and for the comments. Below you will find our responses.

Yours sincerely,

Gerda de Kuijper (on behalf of the authors)

Reviewer 3.

1) I would  have preferred an objective measure of statistical improvement.

We agree with the reviewer, but had to deal with the available files and CCE-policies on the recording of interventions. We mentioned this issue in the limitation section and added that our outcome measure was not validated:

‘First, standardised and validated outcome measures on the effect of interventions are missing and outcomes had to be broadly assessed from information in the files.’

2) I prefer to put the word descriptive study in the title.

We changed the title accordingly:

‘Difficulties in addressing diagnostic, treatment and support needs of individuals with intellectual disability and persistent challenging behaviour: a descriptive file study of referrals to an expertise centre’.

Round 2

Reviewer 1 Report

Accept